# Multiscale structural complexity assessment of coral reefs using underwater photogrammetry

**Erick Barrera-Falcón**[1,2], **Rodolfo Rioja-Nieto**[2☯*], **Roberto C. Hernández-Landa**[3]

**1** Posgrado en Ciencias del Mar y Limnología, Instituto de Ciencias del Mar y Limnología, Universidad Nacional Autónoma de México, Ciudad de México, México, **2** Laboratorio de Análisis Espacial de Zonas Costeras (COSTALAB), Facultad de Ciencias, UMDI-Sisal, UNAM, Mérida, México, **3** Escuela Nacional de Estudios Superiores, Unidad Mérida, UNAM, Mérida, México

☯ These authors contributed equally to this work.
* rrioja@ciencias.unam.mx

## Abstract

Understanding the structural complexity of coral reefs is essential for assessing their condition, biodiversity, and resilience. Traditional methods commonly use a rugosity index based on the chain method, which overlooks the underlying structure of coral reefs. However, digital underwater photogrammetry allows the construction of coral structure models, which can then be used to decompose reef topography across multiple layers. This study introduces a wavelet-based method for the multiscale analysis of reef structural complexity, considering reef's surface and underlying characteristics. Data were collected from six reefs within the Cozumel Reefs National Park (CRNP) at depths ranging from 6 to 14 m. High-resolution Digital Elevation Models (DEMs) and orthomosaics were constructed using digital underwater photogrammetry (UWP). The elevation profiles extracted from the DEMs were analyzed using a Maximum Overlap Discrete Wavelet Transform (MODWT), with a Daubechies mother wavelet to decompose the reef topography into local complexity (related to live coral cover) and underlying complexity (related to the historical context of the formation of the reef matrix). The wavelet-based method effectively decomposed the DEMs into components representing structural complexity at different scales, with the reconstructed DEMs statistically equivalent to the original data source ($p > 0.05$). The underlying reef characteristics contributed the most to the complexity estimates. Significant differences in structural complexity were observed between reefs ($p < 0.05$), where interpretations differed based on the contribution of the surface and underlying characteristics. For the CRNP, Agariciid and branching corals were the primary drivers of surface complexity ($p < 0.05$), rather than mound and boulder and meandroid corals. Our findings indicate that the chain method undervalues the historical role on assessments and the importance of local characteristics in sustaining reef structural complexity over time.

**Data availability statement:** The base dataset required to replicate the results can be found in (https://doi.org/10.6084/m9.figshare.28774106). All other relevant data are within the paper and its Supporting Information files.

**Funding:** RRN was funded by the Programa de Apoyo a Proyectos de Investigación e Inovación Tecnológica, grant number IN218219 from the National Autonomous University of Mexico. EBF was funded with a posgraduate scholarship from the Consejo Nacional de Humanidades Ciencias y Tecnologías.

**Competing interests:** The authors have declared that no competing interests exist

## Introduction

The growth and spatial distribution of scleractinian corals shape coral reef seascapes [1,2]. Geologically, coral reef development depends on a delicate balance between accretion and erosion, which is influenced by both physical and biological factors [3,4]. This dynamic interplay regulates the growth potential of a reef through the net carbonate accretion or erosion of its calcareous matrix [5].

Assessing the structural complexity of coral reefs is crucial for understanding their ecosystem services and processes [1,6]. Structural complexity refers to the physical architecture of a reef, including the organisms that colonize the calcareous matrix and add dimensionality [1,7]. Structural complexity is vital for reef communities and is positively correlated with fundamental biological processes such as feeding relationships, reproduction, and competition [8,9].

Initial attempts to measure reef structural complexity date back to the 1970s [10], when the technique of laying transects with a chain along a single linear profile over a substrate was introduced. This approach led to the development of the rugosity index as a proxy for estimating reef structural complexity [11], which is now commonly referred to as the chain method [12]. Since then, rugosity has become the most widely accepted metric for representing the structural complexity of coral reefs [13,14]. The chain method quantifies the ratio of the distance a chain travels when laid over the substrate to the linear distance between its ends [12]. On flat surfaces this ratio is equal to one, with values greater than one indicating an increased structural complexity [11].

Alternative approaches have been suggested for measuring structural complexity, that incorporate a third dimension [15,16]. These techniques were costly, logistically challenging, and required sophisticated equipment, advanced programming knowledge, and powerful computing resources, making their widespread implementation difficult [15,17]. However, improvements in underwater techniques, particularly in data acquisition and advancements in image analysis, such as structure from motion (SfM) algorithms, have facilitated the integration of a third dimension into measurements [18,19]. Close-range photogrammetry was introduced in coral reef studies to obtain structural complexity metrics for coral colonies with simple morphology [20]. Since then, studies have incorporated photogrammetry, hereafter referred to as underwater digital photogrammetry (UWP), to analyze the structural complexity of coral reefs [21–25]. UWP allows for a more precise assessment of individual colonies over large areas [26,27], making it a new standard for coral reef assessments [28].

The UWP facilitates the utilization of derivative products, such as Digital Elevation Models (DEM) and 3D mesh to investigate the structural complexity of coral reefs. DEMs derived from UWP, have been utilized to develop metrics that can be categorized into three distinct groups: terrain variability metrics (TVM), morphology metrics (MM), and other complexity metrics (OCM) [29]. TVM includes metrics such as linear rugosity, surface rugosity, and surface complexity. MM encompasses metrics like slope, curvature, and the bathymetric position index. OCM comprises metrics such as fractal dimension (D64) and vector roughness measurement (VRM), all of which are employed to assess reef structural complexity [23,30,31]. In this context, the use of

DEMs generated from 3D models has been essential for measuring reef-scale areas, while the application of specific 3D models has been applied to colony-scale analysis with metrics like curvature [32].

Despite advancements in UWP, the integration of multiscale analysis in spatial assessments of coral reefs has not yet been fully explored. Existing metrics, like Vector Roughness Measurement, allow for analysis at varying scales or window sizes, with the results and interpretation being scale-dependent [33]. Other approaches assessing complexity at different scales (e.g. biotope, benthic community, geomorphological zone), commonly adjust the window of analysis or work at different DEM resolutions [34]. Those adequately capture the spatial variability of reef structural complexity [26,34,35]. However, the decomposition of topographic signatures to reveal the underlying structural features across different scales has not been investigated. Therefore, it is important to develop a robust and scalable technique that can enhance our understanding of the complex topography of reefs and their spatial heterogeneity. By addressing the limitations of current methods and focusing on distinguishing historical accretion from live coral conditions, these techniques can provide clearer insights that are essential for ecological assessments and long-term conservation strategies.

This study proposes the application of a wavelet filter to high-resolution coral reef elevation profiles obtained using UWP to analyze coral reef topography and characterize reef structural complexity. By conducting a spatial decomposition of topographic signatures, we facilitated a comprehensive multiscale analysis [36–39]. Specifically, this study aims to (1) use a wavelet filter on high-resolution coral reef elevation profiles of six shallow reef systems of Cozumel Island and (2) conduct a spatial decomposition of topographic signatures to enable a multiscale analysis.

## Materials and methods

### Study area

The Cozumel Reefs National Park is located in the municipality of Cozumel, 16.5 km off the coast of the state of Quintana Roo (Fig 1). Cozumel is the second most populated island in Mexico, with c.a. 84500, inhabitants [40]. The main economic activity is tourism, which is associated with SCUBA diving and cruise ships [41]. Since the 1970s, population density and urban development along the coast have increased considerably, leading to significant changes in land use and modifications to wetland ecosystems along the western margin [42].

Marine habitats in the CRNP include a mix of marginal reefs, patch reefs, and mixed corals over hard calcareous substrates, along with algal meadows, seagrass beds, and mangrove areas [43]. The shallow sublittoral slope tends to be narrow and descends gradually from the coast, with the most developed reefs found along the edge of the southwestern insular platform [43]. Most of the population is located in San Miguel town, with tourism infrastructure developed in the Northwest and Southwest areas of the island. The reefs selected for this study are distributed along a north-south gradient, characterized by an increase in coral cover, reef development, and the intensity of tourism-related activities in the same direction. [44]. The region is affected by hurricanes [45,46], and corals are threatened by emerging diseases such as the Stony Coral Tissue Loss Disease [47] and the massive influx of *Sargassum* spp. [48].

### Data collection

Data were collected from six reefs within the CRNP at depths ranging from 6 to 14 m (Fig 1). Three plots measuring 5×30 m were established at each reef, covering a total area of 450 m² per reef. The plots were delineated with markers made of 60×60 cm polyvinyl chloride (PVC) quadrats, placed at each corner and at the midpoint of each plot. Divers swam (along and across the plots) at c.a. 5 m/s, maintaining 2 m from the surface, capturing photographs with Canon G12 cameras, ensuring a high degree of overlap between adjacent images to facilitate accurate photogrammetric reconstruction [49]. Depth was recorded at the central point of each marker to provide precise elevation references.

No organisms were collected for this study, and authorization (No. DPNAC/311/2019) from the CRNP authorities was obtained to perform fieldwork activities.

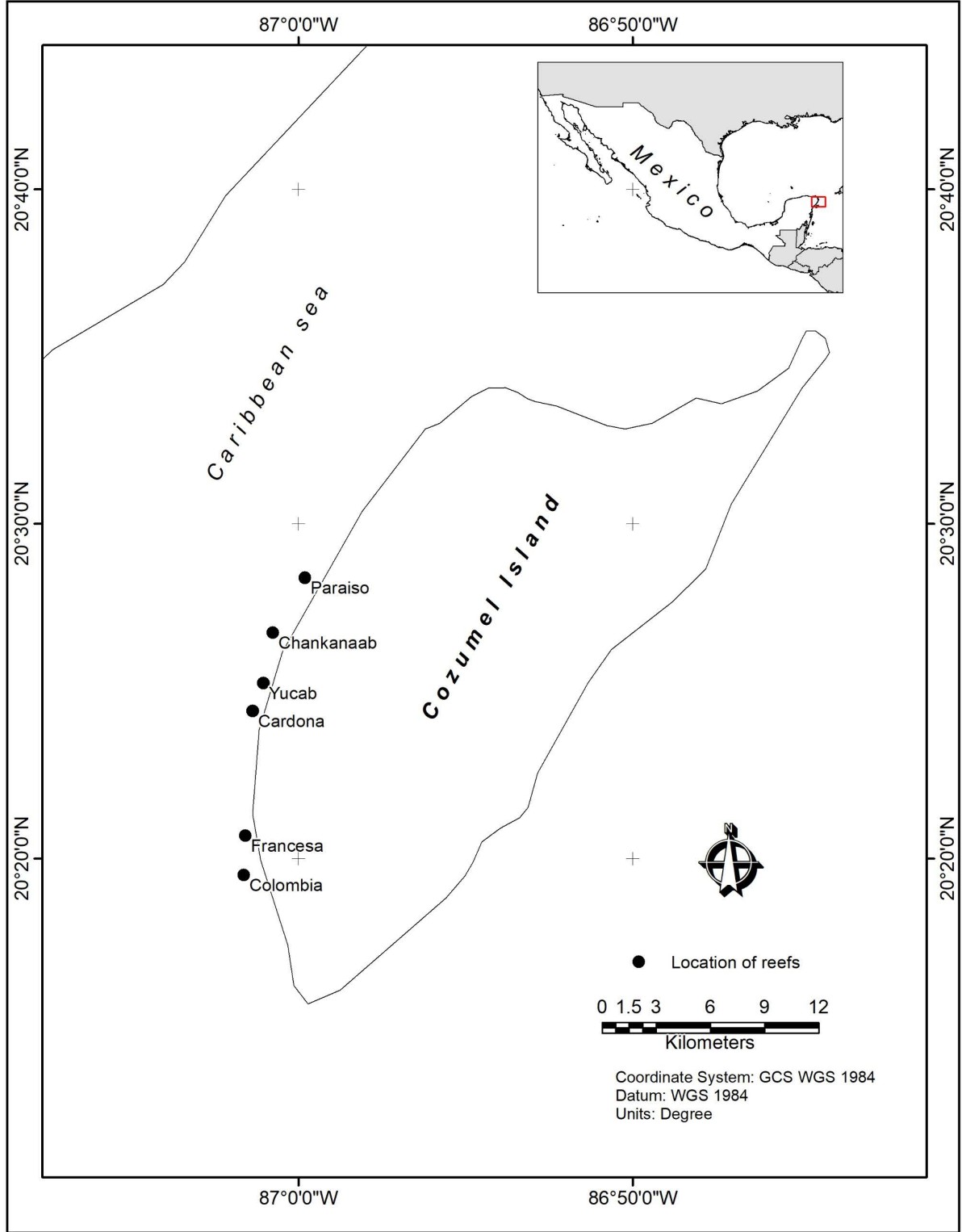

**Fig 1. Study Area, black dots indicate the locations of the studied reefs within the Cozumel Reefs National Park.** Vector files used to construct this map were obtained from Natural Earth under a CC0 1.0 Public Domain license.

## Photogrammetric processing, colony identification and digitization

The images from each plot were processed to construct digital elevation models (DEMs) and orthomosaics using Agisoft Metashape software v1.5. The main workflow involved aligning the images, identifying the control points (60×60 cm PVC markers), and applying scale corrections using known dimensions of the markers. The parameters used for alignment were high accuracy, type estimated, key point limit set to 40,000, and tie point limit set to 10,000. Re-optimization of the alignment was performed to correct horizontal discrepancies, and the Z-axis was fixed by incorporating the depth data recorded from the field markers to ensure accurate vertical scaling [49]. The obtained scale and root mean square errors were mostly below 0.02 m and 0.35 m, respectively (S1 File). A dense point cloud was generated using the corrected tie points (source: deep maps at high level), which was then used to extract the DEMs and build an orthomosaic for each plot. The files were standardized to a spatial resolution of 0.008 m for the DEMs and 0.006 m for orthomosaics. The DEMs were exported in raster format by assigning an arbitrary projected coordinate system to facilitate the spatial analysis (EPSG:32616).

The orthomosaics were analyzed in ArcMap v. 10.8 [50] where coral colonies were identified, digitized, coded, and grouped into morphological groups (S2 File) according to the Atlantic and Gulf Rapid Reef Assessment (AGRRA) protocols [51]. Species identification was performed visually using the Reef Coral Identification Guide [52], Coralpedia (https://coralpedia.bio.warwick.ac.uk/), and AGRRA identification guides [51], considering colonies ≥ 5 cm, which is the minimum size that can be identified in the orthomosaics [49].

## Assessment of rugosity and structural complexity

*In situ* rugosity was assessed in each plot using the chain method [53], which involves laying a chain along the contour of the substrate over a 10-meter transect and calculating the ratio of the chain length to the linear distance.

Digital structural complexity was assessed in the laboratory using algorithms written in MATLAB v2020 (available at https://github.com/Ebfsig/Multiscale-Rugosity-). The Maximum Overlap Discrete Wavelet Transform function (MODWT) with the Daubechies as the mother wavelet [54] was applied to height values, treated as time-series data, using each row of pixels from the DEMs to decompose the signal into approximation and detailed coefficients across four levels [55,56]. This number of levels is commonly used in this type of analysis and using an increased number results in very small variations that are not represented in the DEMs spatial resolution [57,58].

The process involved convolving the signal with low-pass and high-pass filters at each level to obtain the coefficients that were then used in the next level of analysis.

The original signal $X_t$ was then reconstructed using the coarsest approximation $\widetilde{V_4}$ and all detailed coefficients W1 to W3 obtained from each level (equation 1). This allowed to capture both large scale trends and fine details in the DEMs characteristics in a multiscale analysis of surface features.

$$X_t = \widetilde{V_4} + \sum_{j=1}^{4} \widetilde{W_j}$$

(1)

To evaluate DEM decomposition, the original model was compared with the sum of the topographic signatures using the Mann-Whitney U test (p = 0.05). The sum of the first four decomposition levels was considered as inherent surface variations (e.g., those determined by coral colonies), representing small-scale features, hereafter defined as digital local complexity (DLC). The approximation at level five was defined as the underlying structure, representing larger-scale relief without inherent variations, defined as the digital global complexity (DGC). This approach is analogous to creating a digital terrain model that excludes surface roughness and focuses solely on a broader underlying matrix.

For each profile, the digital structural complexity was estimated using the roughness parameter Ra [36]. Ra, represents the arithmetic mean of the absolute deviations of the profile height from the mean line over a specified evaluation length (5 cm), calculated using equation 2.

To standardize the digitization process and extract precise measurements of Ra, an evaluation length of five centimeters was used to assess both the DLC and DGC profiles. This length corresponds to the minimum diameter of the digitized coral colonies.

$$Ra = \frac{1}{L} \int_0^L |h| dx$$

(2)

Where:

h = absolute height of the surface profile at each sampling pixel.

L = evaluation length.

The results were stored in the same spatial position as the corresponding pixels, resulting in a raster that displayed spatial variations in reef structural complexity.

## DLC and DGC based on coral cover

The vector files of the digitized coral colonies were rasterized, with each pixel tagged with a species code to enable spatial filtering of the data. Rasterization considers the pixel size of the DLC and DGC matching the pixel size of the DEMs. The datasets were organized into a three-dimensional matrix. The matrices representing the DLC and DGC occupied the first and second dimensions, respectively, whereas the raster of coral cover was placed in the third dimension.

To analyze the data and decrease the processing time, we considered 380 m² per reef, as this is representative of the coral community of the shallow reefs in the study area [49]. Spatial overlap was performed by filtering the DLC and DGC matrix values according to the site and coral groups. To compare the *in situ* rugosity with the total complexity (DLC+DGC), we considered the 380 m². As the data were not normally distributed, comparisons of digital complexity among reefs and the contributions of coral shape groups were conducted using the non-parametric Kruskal-Wallis test (H test, p = 0.05). When the test indicated significant differences, pairwise comparisons between sites were performed using Dunn's post-hoc test with Bonferroni correction [59].

A linear regression was used to evaluate the relationship between coral cover, DLC, and DGC. Coral cover (m²) per reef was calculated using Arcmap v10.8 [50], and the data were then normalized by applying a log transformation to the area. The analysis was performed using R base 4.2.2 [60].

## Results

The reconstruction using the wavelet approach and the decomposition in DLC and DGC for Colombia and Chankanaab reefs are shown in Fig 2 as an example (for the other sites, see S3 File). In all cases, the reconstructed DEMs were equivalent to the original DEM (p > 0.05; S4 File), indicating that wavelet filtering effectively decomposes the DEM for complexity analysis in both the DLC and DGC. Furthermore, the DGC was higher than the DLC in all cases (Fig 2, S3 File).

*In situ*, the rugosity measured with the chain method showed that Francesa and Colombia had higher values than Yucab, Chankanaab, Cardona, and Paraiso, where the latter showed the lowest values (Fig 3). For total digital complexity (DGC+DLC), considering 380 m² of analysis, all sites showed significant differences (p < 0.05, S5 File). The site Yucab exhibited the highest values, followed by Colombia, Francesa, Cardona, Chankanaab, and Paraiso (Fig 3). No correlation was observed between *in situ* rugosity and total digital complexity (R² = 0.14, p > 0.05).

When assessing DGC exclusively (Fig 4) in areas underneath live coral cover, Yucab, Francesa, and Colombia reefs had the highest values, whereas Paraíso showed the lowest structural complexity; all sites were statistically different (p < 0.05, S6 File).

For DLC (Fig 5), Colombia, Francesa, Cardona, and Paraiso showed the highest values, whereas Yucab exhibited the lowest values. All sites showed significant differences (p < 0.05; S7 File).

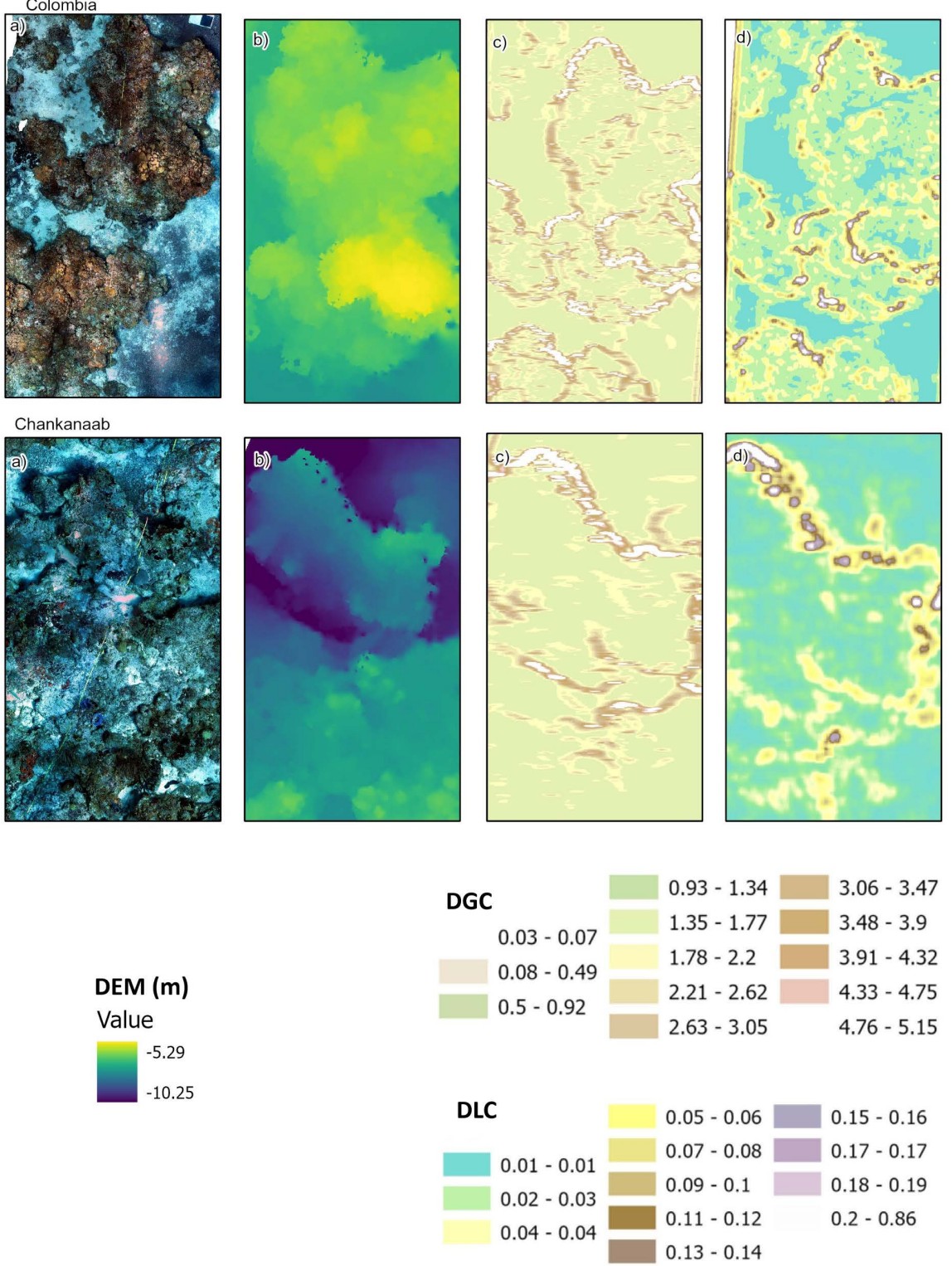

**Fig 2. Digital complexity values for Colombia and Chankanaab reefs.** Results for the other reefs are presented in the S3 File. (a) Orthomosaic, (b) DEM (Digital elevation model), (c) DGC (digital global complexity) highlighting the historical contributions to reef structure, and (d) DLC (digital local complexity), showing the biological contributions from live coral communities.

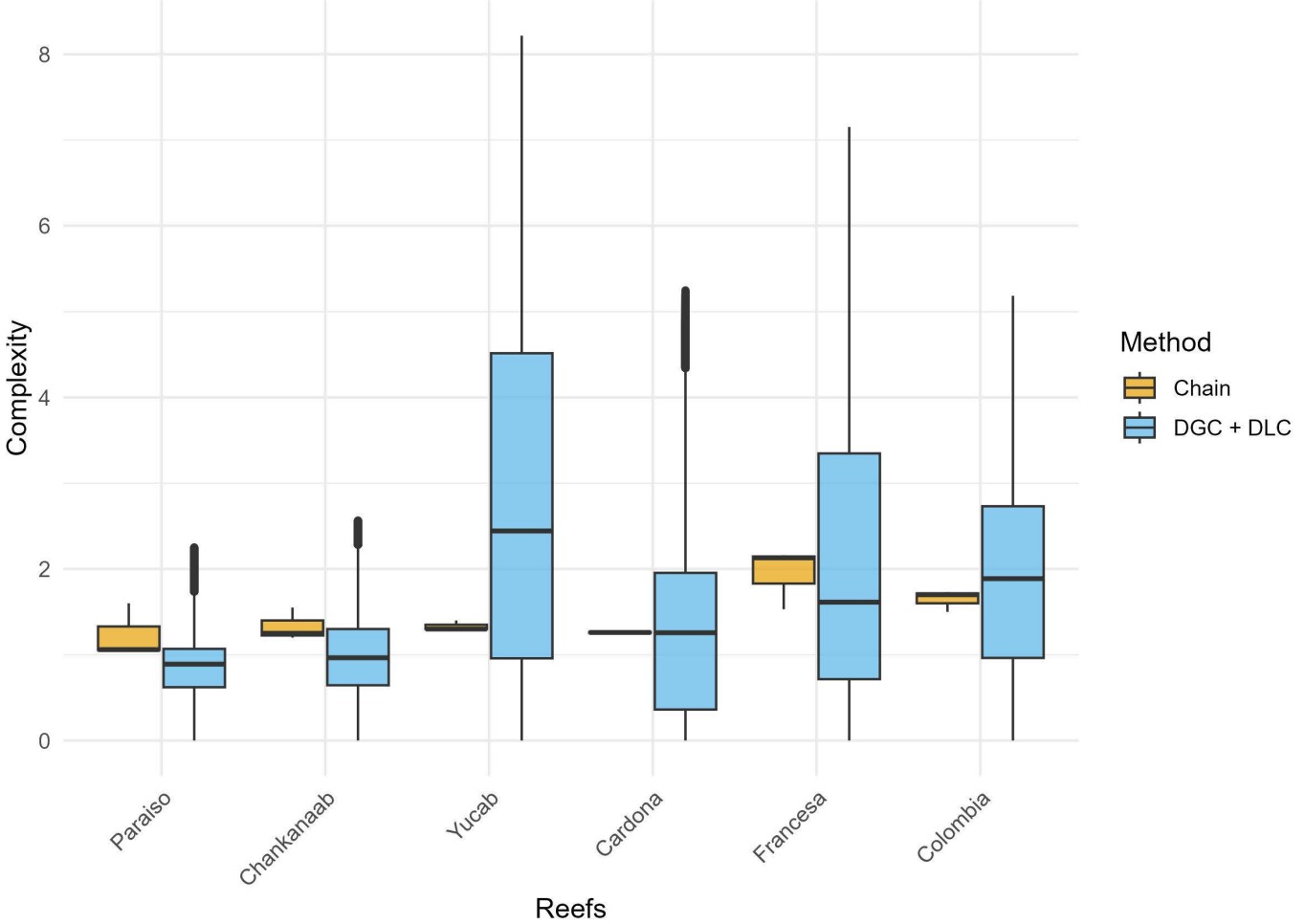

**Fig 3. In situ (chain method), and total digital complexity (DGC + DLC) per reef.** The ordering of reefs follows a north-to-south direction. The black lines represent the median values, highlighting the variation among reefs. The total digital complexity considers 380 m² of analysis.

Colombia and Francesa reefs had the highest complexity in both the DGC and DLC estimates. However, the results for Yucab were contrasting, as this reef presented the highest DGC but the lowest DLC values.

In terms of contribution to DLC by coral morphological groups in the CRNP, the agariciid and branching groups were the highest contributors ($p < 0.05$) (Fig 6). No significant differences were observed between the flower and solitary, Meandroid, and Mound and boulder morphological groups ($p > 0.05$, S8 File).

The DLC shows a significant linear correlation with total coral cover ($R^2 = 0.63$, $p = 0.017$). No relationship ($p > 0.05$) was observed between coral cover and DGC (S9 File). When assessing the linear relationship between DLC and coral morphological groups (Fig 7), a significant positive correlation was identified for the Agariciids ($R^2 = 0.88$, $p = 0.006$), and a marginal correlation was observed for the Mound and Boulder species ($R^2 = 0.60$, $p = 0.069$). No other significant correlations were detected ($p > 0.05$). The DLC for the reefs evaluated in the CRNP is driven mainly by the Agariciids group.

The Agaricia spp., Porites spp., Orbicella spp., Siderastraea spp., and *Montastraea cavernosa* were the species with the highest cover, which belong to agariciid and massive groups (Fig 8). A detailed description of the species contribution to coral cover for these reefs can be found in [47].

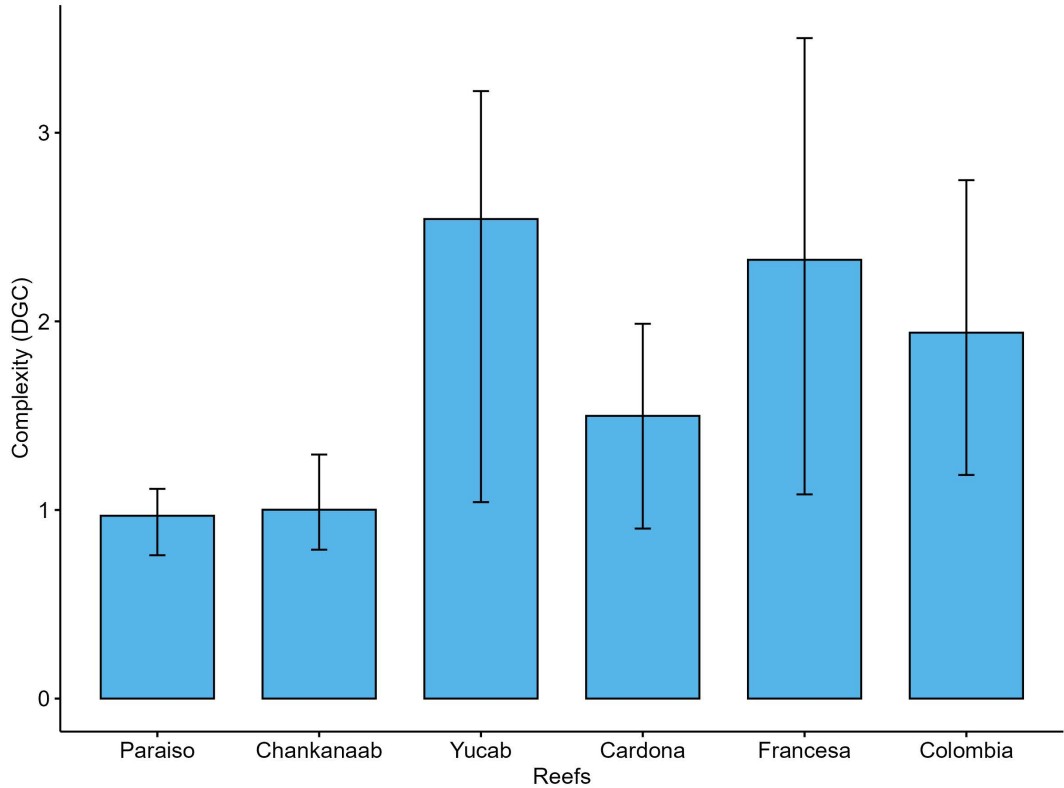

**Fig 4. DGC values per reef.** DGC was calculated only beneath the areas where live corals were present. The ordering of reefs follows a north-to-south direction. The columns represent the median values, and the bars indicate the standard error.

## Discussion

Underwater digital photogrammetry improves the assessment of coral reef features by allowing more accurate quantification of structural complexity, which is a key factor in understanding ecological processes and biodiversity patterns [24,27,61,62].

The application of wavelet based multiresolution analysis, which decomposes complex data into various levels of resolution, has demonstrated efficacy in other scientific fields, such as the analysis of soil particle sizes from digital images [63] and the assessment of bone structures [64]. In coral reefs, wavelets have been applied to image analysis to capture fine details and textures, making them well-suited for coral classification or detection of marine species such as the crown of thorns starfish [65,66].

We used Digital Elevation Models (DEMs) to accurately represent reef structural complexity across different scales. The statistical equivalence between the reconstructed and original DEMs confirmed the reliability of our approach. Discrepancies emerged when comparing the assessments obtained from the *in situ* and digital methods.

The *in situ* approach identified the Francesa, Colombia, and Yucab reefs with the highest complexity. In contrast, the total digital complexity approach highlighted Yucab, Colombia, and Francesa reefs with the highest values. The chain method relies on physical measurements along specific transects, and its accuracy depends heavily on the number of replicates and the exact sites surveyed [67,68]. Limited sampling can lead to the under-representation of the spatial variability of the reef, potentially biasing the results if the surveyed areas are not representative of the entire reef [14,69,70]. This is not the case when using the approach based on UWP, as large areas (in this case 380 m²) are considered for the assessment.

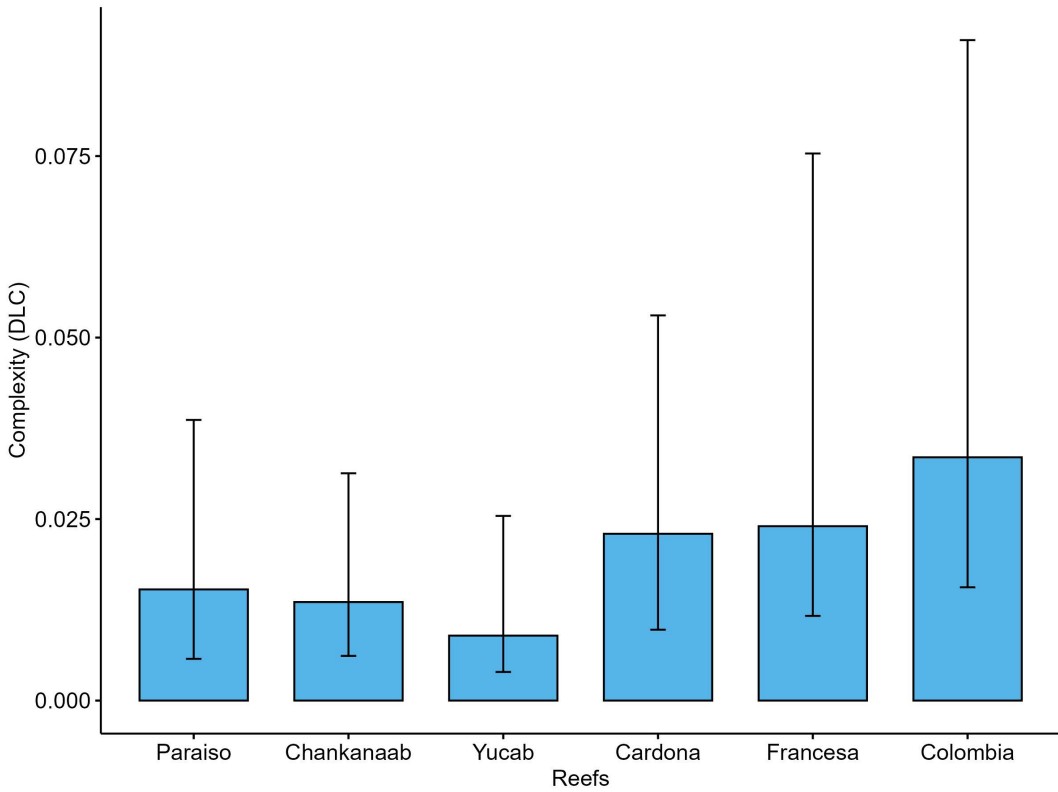

**Fig 5. DLC values per reef, which is the structural complexity contributed by live coral cover.** The ordering of reefs follows a north-to-south direction. The columns represent the median values, and the bars indicate the standard error.

Furthermore, our approach decomposes the reef structural complexity assessment at multiple spatial scales, separating the historical contribution from the complexity related to the live hard coral cover. Hard corals are the main contributors to reef structural complexity [71]; therefore, we integrated the dynamic capacity of organisms to maintain reef structural complexity. This is essential for accurately assessing reef condition, ecological functioning, and resilience.

While various complexity metrics like fractal dimension, VRM, multiscale roughness, slope, aspect, and profile curvature can be derived from UWP-generated DEMs, with the exception of fractal dimension the metrics are influenced by the analysis's scale or orientation [29,72]. Furthermore, when analyzing structural complexity at different scales, it's common to choose different source resolutions or adjust the windows of analysis [34,73,74]. In contrast, the wavelets method offers a distinct advantage as it doesn't require an analysis window to deconstruct the topographic signal. Instead, it breaks down the topographic signal into two key components.

Significant differences in complexity exist among the reefs evaluated, driven by both the DLC and DGC. This complexity is shaped by benthic components, such as live coral cover and the calcareous matrix of the reef structure, collectively contributing to the total complexity. Digital underwater photogrammetry allows for a more precise quantification of structural complexity [14,27,28], which is crucial for gaining a deeper understanding of the ecological processes within these habitats, such as the preference of refuge for small-bodied schooling fishes [75], and the effect of structural complexity on coral recruitment [76].

A clear distinction emerges in the interpretation of structural complexity at both the DLC and DGC levels, consistent with findings from previous studies that revealed that different spatial scales yield different patterns among sites [68], and

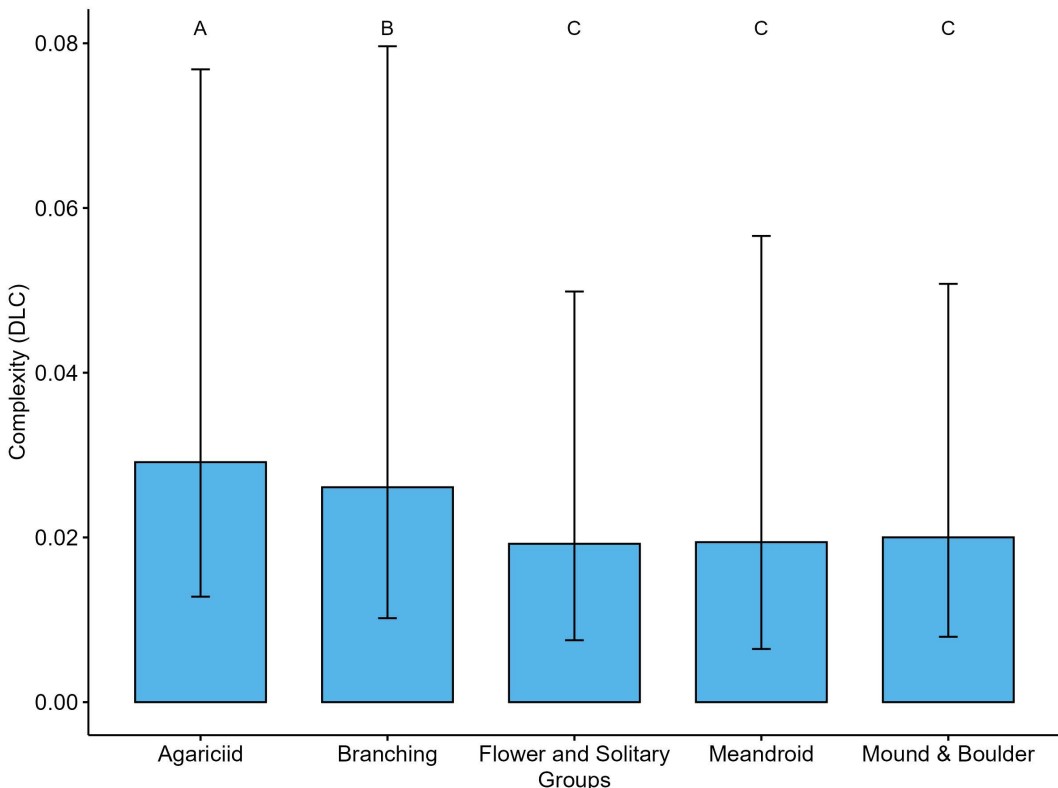

**Fig 6. Contribution of coral groups to DLC in the CRNP.** A and B indicate statistically significant differences (p < 0.05) while C shows no significant differences (p > 0.05), among coral groups.

that structural complexity in coral reefs arises primarily from two sources: biotic components such as hard corals and sponges, and coarse-scale topographic variations reflecting the geological context of the reefs [35].

In our study, the DLC was higher in Colombia, Francesa, and Cardona reefs, and the lowest values were observed in Yucab. The DGC was higher in Yucab, Francesa, and Colombia. The difference in the estimates for the Yucab reef is likely related to the contribution from its calcareous matrix formed by the historical accretion of scleractinian corals, reflecting the geological context [35,68]. Yucab has the lowest coral cover (< 4%) compared to Colombia (c.a. 15%), Francesa (c.a. 4.55%) and Cardona (5.51%). A dramatic decline in coral cover between 2005 and 2008 was observed in the reefs of Cozumel, and this decrease was particularly notable in Yucab, where the coral cover dropped by approximately 77% [77,78]. Despite the considerable loss, Yucab maintains high DGC values, suggesting that this site was once highly complex.

When focusing on DLC based solely on sites with coral cover, the highest values were observed in Colombia, which also had the greatest coral cover [49]. The DLC is primarily defined by corals with high cover, with the dominant groups being agariciid corals, followed by branching corals [49]. The correlation between local complexity and coral cover suggests that contributions to reef DLC are closely linked to species dominance.

Agariciid and branching corals, especially *Agaricia agaricites* and *Porites porites*, currently drive the DLC in the evaluated reefs. The former are regarded as opportunistic "weedy" species [9,79], which rapidly colonize disturbed areas and can dominate coral assemblages when competition for space is reduced [80]. These morphological groups are characterized by low relief, foliose, and ramified morphologies, which have historically made a limited contribution to the calcareous matrix [79,81]. This highlights a potential imbalance; while these groups contribute significantly to DLC, they offer limited

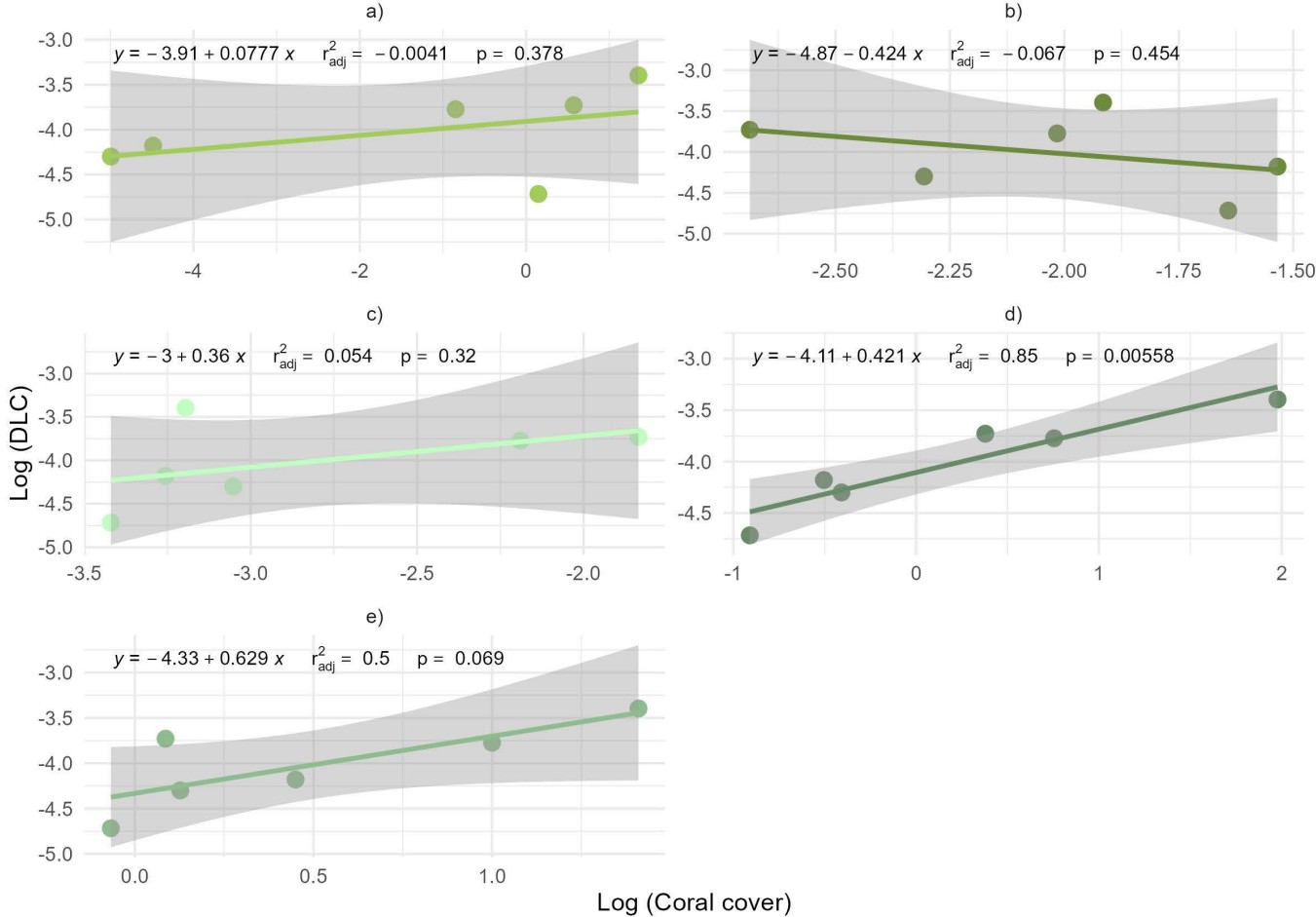

**Fig 7. DLC and coral cover grouped by morphological groups.** a) Branching, b) Meandroid, c) Flower and Solitary, d) Agariciid) and e) Mound and Boulder.

structural support [79], potentially compromising the long term physical functionality of reefs. Moreover, the scarcity of mound and boulder species essential for supporting long-term structural complexity [82,83] emphasizes the need to evaluate reef health beyond coral cover.

Concentrating exclusively on overall complexity (either total digital complexity or the chain method) to assess reef condition can lead to an underestimation of the capacity of the reef to maintain structural complexity and resilience over time. Our findings reveal that the DGC, which provides a historical context, is the main factor influencing complexity estimates, and this needs to be further assessed.

Our results underscore the spatial variability of reef structural complexity within the CRNP. The elevated complexity values observed in Yucab and Francesa suggest that these reefs may offer more intricate habitats and provide greater refuge for marine organisms [24,84]. In contrast, the reduced complexity in Paraíso indicates a flatter reef structure, potentially influencing the species inhabiting these areas [2]. Evaluating coral cover and its contribution to DLC, alongside focusing on the principal groups that enhance DLC, can facilitate the identification of sites that may support positive reef growth [3]. By mapping both DLC and DGC, assessing coral species composition, and identifying sites with historically high complexity but low local biological complexity, conservation efforts can be targeted more effectively. Specifically,

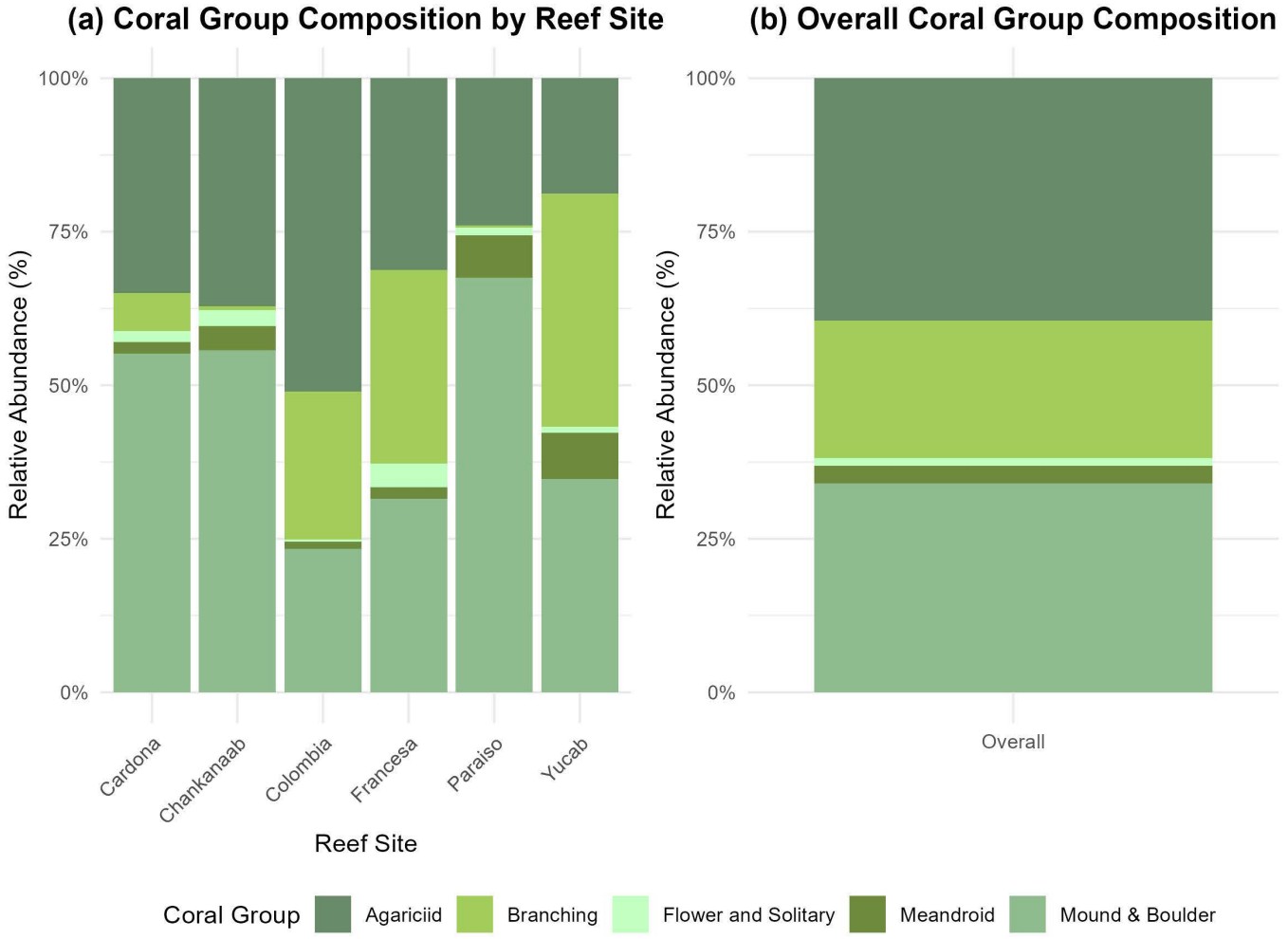

**Fig 8. Relative coral cover for each reef: a) species cover grouped by morphological groups per reef and b) morphological groups coverage across all reefs.** The species are grouped in morphological categories according to the AGRRA protocol, with detailed information about species and groups available in S2 File.

efforts can be concentrated on enhancing local complexity through restoration, safeguarding existing structural foundations, and monitoring changes in reef topography and biological composition over time.

## Conclusions

This study illustrates the effectiveness of a wavelet-based approach for separating high-resolution digital elevation models (DEMs) of coral reefs into detailed digital local complexity (DLC) and digital global complexity (DGC) components, enabling a comprehensive multiscale examination of reef structural complexity. By employing a wavelet filter on elevation profiles extracted from UWP derived DEMs, we were able to capture both fine-scale (DLC) and broad-scale (DGC) topographic features, providing a better understanding of reef structural complexity.

Our results revealed that agariciid and branching corals are the main contributors to DLC in the CRNP. Although these "weedy" species can rapidly colonize disturbed areas, their prevalence may not support long-term structural complexity and could indicate a degraded ecosystem. The lack of crucial coral groups necessary for maintaining long-term structural integrity underscores that coral cover alone is insufficient to evaluate reef health.

Reef areas exhibiting high DGC but low DLC, such as Yucab, may appear structurally complex but lack the living coral cover required to sustain diverse marine communities and maintain vital ecosystem functions. Therefore, conservation initiatives should aim not only to preserve the physical structure of reefs but also to improve local complexity through the restoration and protection of coral species that contribute significantly to DLC.

A primary limitation of our methodology is the use of 2D format DEMs, which may not fully capture the intricacies of certain coral colonies, particularly those that grow in clumps. UWP provides information visible from the perspective of a camera, potentially overlooking complex three-dimensional structures. The resolution of DEMs can constrain the obtention of detailed features. Nonetheless, the resolution of our data is adequate for assessing complexity at the colony level, thereby enabling us to examine potential long-term implications. Future studies should investigate the application of this methodology to 3D models to represent the full complexity of coral reef structures.

## Supporting information

**S1 File  Metashape and dataset parameters.**
(DOCX)

**S2 File  List of species identified, coded and classified based on coral morphology.**
(DOCX)

**S3 File  Digital complexity values for each reef.**
(DOCX)

**S4 File.  Statistical analysis comparing DEM versus DEM using wavelet reconstruction.**
(DOCX)

**S5 File.  Statistical analysis of digital complexity for each reef.**
(DOCX)

**S6 File.  Statistical analysis of DGC for each reef.**
(DOCX)

**S7 File.  Statistical analysis of DLC for each reef.**
(DOCX)

**S8 File.  Statistical analysis of DLC in comparison to Group.**
(DOCX)

**S9 File.  Correlation DLC, DGC versus coral cover.**
(TIF)

## Author contributions

**Conceptualization:** Rodolfo Rioja-Nieto.

**Data curation:** Erick Barrera-Falcón, Rodolfo Rioja-Nieto, Roberto C. Hernández-Landa.

**Formal analysis:** Erick Barrera-Falcón.

**Funding acquisition:** Rodolfo Rioja-Nieto.

**Investigation:** Erick Barrera-Falcón.

**Methodology:** Erick Barrera-Falcón.

**Project administration:** Rodolfo Rioja-Nieto.

Resources: Rodolfo Rioja-Nieto.

Software: Erick Barrera-Falcón.

Supervision: Rodolfo Rioja-Nieto.

Validation: Erick Barrera-Falcón, Roberto C. Hernández-Landa.

Visualization: Erick Barrera-Falcón.

Writing – original draft: Erick Barrera-Falcón, Roberto C. Hernández-Landa.

Writing – review & editing: Erick Barrera-Falcón, Rodolfo Rioja-Nieto, Roberto C. Hernández-Landa.

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
