## [Decision Letter · Decision Letter 0]

Dear Dr. Rioja-Nieto,

Thank you for submitting your manuscript to PLOS ONE. After careful consideration, we feel that it has merit but does not fully meet PLOS ONE’s publication criteria as it currently stands. Therefore, we invite you to submit a revised version of the manuscript that addresses the points raised during the review process.

We look forward to receiving your revised manuscript.

Kind regards,

Parviz Tavakoli-Kolour

Academic Editor

PLOS ONE

Journal Requirements:

“This research was funded by the PAPIIT grant (IN218219) from the National Autonomous University of Mexico supported this work. EB-F was supported by the CONACYT postgraduate scholarship.”

“RRN was funded by the Programa de Apoyo a Proyectos de Investigación e Inovación Tecnológica, grant number IN218219 from the National Autonomous University of Mexico. EBF was funded with a posgraduate scholarship from the Consejo Nacional de Humanidades Ciencias y Tecnologías.”

6. Please include captions for your Supporting Information files at the end of your manuscript, and update any in-text citations to match accordingly. Please see our Supporting Information guidelines for more information: http://journals.plos.org/plosone/s/supporting-information .

7. We are unable to open your Supporting Information file [zip file]. Please kindly revise as necessary and re-upload.

Reviewers' comments:

Reviewer's Responses to Questions

**Comments to the Author**

1. Is the manuscript technically sound, and do the data support the conclusions?

Reviewer #1: Yes

Reviewer #2: No

2. Has the statistical analysis been performed appropriately and rigorously?

Reviewer #1: Yes

Reviewer #2: Yes

3. Have the authors made all data underlying the findings in their manuscript fully available?

Reviewer #1: Yes

Reviewer #2: No

4. Is the manuscript presented in an intelligible fashion and written in standard English?

Reviewer #1: Yes

Reviewer #2: No

Reviewer #1: Overall

Some references are in a different format (numbered vs. spelled out), but maybe this is the journal instructions. They should be validated if they are appropriate.

Introduction Section

The introduction seems to miss the existing literature on this topic. I recommend more thorough literature search on the topic of coral reef photogrammetry and rugosity. In particular, see Carlot et al 2020 (https://doi.org/10.1007/s00338-020-01916-8); Fukunaga et al 2019 (https://doi.org/10.3390/jmse7020027); Fukunaga et al 2020 (https://doi.org/10.3390/rs12061011); Fukunaga & Burns 2020 (https://doi.org/10.3390/rs12172676) to list a few. It is acknowledged this is not a review paper, but having an updated background pertinent to your topic is useful in pushing this field further.

M&M Section

Photogrammetric processing, colony identification and digitization

General – More information should be included on the Metashape parameters used to create the orthomosaics and DEMs. What were the alignment parameters and error reduction protocols used to generate the outputs? Additional, what is the accuracy of the 60x60 cm PVC markers used to scale the plots? Markers used for photogrammetry tend to be high resolution in order to provide appropriate level accuracy of the constructed models. What is the resolution of the models, what resolution were the orthomosaics, and DEMs exported. The stated protocol was a 2m altitude which is a bit high for high resolution, but helps cover larger areas, but also tends to decrease the output resolution. All this information should be incorporated somewhere in the manuscript. Also, see the Metashape report metrics which also provides additional contextual information on the models.

I recommend you provide the parameters and report metrics as well as the root mean square error of all the models as supplementary files. This provides a background on the models generated and their outputs.

Assessment of Rugosity

Line 145 – acronym is incorrect, should be DLR

General – This seems to be a fairly novel approach in assessing photogrammetry models developed in other disciplines. As stated in my comments about missing relevant research in looking at similar questions about coral reefs and community structure, I suggest you incorporate some comparison analyses at some level in order to help support the use of this novel approach. It may be appropriate, but without a comparison to existing published methods, it might provide similar or different results to these methods. This approach does appear to be different by separating out local vs global rugosity, so this is certainly worth pursuing and it well supported. I recommend additional methodical description on the MODWT and Daubechies method used here as it appears to be a novel application. More information on the decomposition levels also seems warranted. Why do the first four levels represent the local features and the final level the global features?

Results Section

Line 220 – should be mound, not mount?

General – See above comments on additional supplemental material.

I am having a hard time going between the results section with the figure captions and the figure themselves. There appears to be some figure reference mistakes as well. I recommend to rework this section and double check figure references. I would be interested in more in-situ vs digital rugosity comparisons.

The other aspect I realize is that I am uncertain of the rugosity metric from the digitized method. I presume it is not the same as the chain ratio method, but what units are this metric in? Unitless? Conducting a digital measurement of rugosity that is similar to the chain method is very interesting. This is a study I have been interested in and this is the first attempt at it. I do recommend flushing that out as much as possible. Even if the two metrics are different, it could be interesting to run a regression on the methods to show the correlation.

Discussion

See comments from the introduction to incorporate the newest literature on this topic. Your discuss is good, but seems to lack an in depth background on the topic. These results, while important, are not new in the global sense. They are worthy of publication, but also should incorporate a discussion of these results from results of other published work based on the wealth of background literature that exists.

Reviewer #2: This study attempts to apply a wavelet-based method using a MATLAB-based algorithm to analyze two different scales of rugosity, DLR and DGR, from DEMs generated through photogrammetry. The research was conducted at 6 reef sites within the CRNP in Mexico, with three 450-square-meter areas analyzed per reef. Non-parametric statistical methods were used to compare rugosity variations among the 6 reef sites and across different coral growth forms. The results suggest that the wavelet-based method provides a more comprehensive understanding of reef rugosity than the traditional chain method. The study claims that DLR reflects the biological contribution of coral growth forms, while DGR represents the topographic variation of the reef (historical development). Among all coral growth forms, Agariciid and branching corals were found to contribute the most to DLR.

I find the application of a method traditionally used for time-series analysis to DEM data to be innovative. The study's effort to integrate multiscale structural complexity into habitat assessment is valuable, and the literature review on the chain method in the introduction is thorough. However, there are several areas where the manuscript could be improved:

Major comment

1. The limitations of the chain method and multiscale structural complexity were discussed as early as 2012 (Friedman et al., 2012), and thus, this is not a novel finding. The main contribution of this study lies in applying a wavelet-based method to analyze rugosity. However, the methodological approach and visualizations do not clearly illustrate the differences between the wavelet-based method and the chain method:

• Methodology (Lines 90-96): Since the results primarily rely on comparisons across different reefs, it is essential to provide a basic description of the 6 reef sites, especially regarding factors that influence structural complexity, such as the anthropogenic activities, historical thermal stress, and high range. This would enhance the interpretation of the rugosity differences across sites.

• Discussion (Lines 235-242): This section should provide a more detailed discussion on how the newly introduced rugosity metrics (DLR and DGR) differ from chain method. Specifically, what aspects of biological or abiotic reef structure do these differences reflect?

• Ecological Relevance (Lines 243-246): Given that the study analyzed coral cover, it would be beneficial to present the relationship between DLR, DGR, and coral cover. This would establish a stronger ecological connection between the proposed rugosity metrics and coral reefs, and allow for comparisons with other existing structural complexity metrics.

• Figures and Statistical Consistency (Figures 3-6): I recommend merging Figures 3-6 to facilitate a direct comparison between the wavelet-based method and the chain method. The statistical approaches and labeling should also be consistent across figures. According to Line 194, Figure 3 presents median values, while other figures use mean values. The rationale for this inconsistency is unclear and should be justified or standardized.

2. The explanation of the rugosity calculation method is insufficient, and the ecological significance of different rugosity scales is not well supported by the results:

• Methodology (Lines 117-118): The quality of photogrammetry images and the resolution of the DEMs significantly impact structural complexity measurements. The manuscript should provide the number of images used and the resolution of the output DEM. Additionally, it would be beneficial to include the parameter settings used in Agisoft Metashape as supplementary material to allow other researchers to compare methodologies.

• Results (Figures 2a, 2b): Presenting all reefs together makes it difficult to observe finer details. I strongly suggest displaying only representative reef sites to enhance clarity. Furthermore, corresponding orthomosaics and DEMs should be included to allow for a direct comparison between DLR, DGR values, and actual ecological features.

• Statistical Interpretation (Figure 7): The significant differences between coral morphological groups in Figure 7 only indicate that DLR can distinguish between Agariciid, branching corals, and other coral growth forms, but this does not necessarily mean they are the primary contributors to reef structural complexity. Since coral morphology data were analyzed in ArcMap, the study should provide the composition of coral growth forms and coral cover at each reef site to better understand which growth forms contribute the most to structural complexity.

• Quantifying Coral Contributions (Line 268): When discussing “…Agariciid corals, which currently contribute the most to DLR,…” it would be more rigorous to use a Linear Model with rugosity log-transformed to quantitatively assess the contribution of different coral growth forms to DLR rather than relying solely on descriptive statistics.

• Missing Data on Coral Cover Relationships (Lines 264-267): The manuscript frequently discusses the relationship between DLR, DGR, and coral cover, but there is a lack of supporting visual data. Although the methodology mentions such an analysis, no figures or correlation analyses are provided. I recommend including scatter plots of DLR versus coral cover at each plot and conducting a correlation analysis.

• Supporting Evidence for Yucab Reef Example (Lines 259-261, 274-277): The manuscript repeatedly states that DLR reflects biological contributions from coral growth forms, while DGR represents the broader topographic variability shaped by historical reef development, using Yucab Reef as an example of high DGR and low DLR. While this is a valuable insight, it requires additional supporting literature or actual reef site photographs to substantiate the claim.

3. The manuscript lacks comparisons with recent studies (post-2015) on reef structural complexity, making it difficult to clearly assess the advantages and limitations of the wavelet-based method relative to other approaches. Without these comparisons, the study does not fully demonstrate how the wavelet-based method differs from or improves upon existing techniques:

• Introduction (Lines 72-74): The manuscript states, “…Existing methods neither adequately capture the spatial variability of reef complexity nor effectively decompose topographic signatures to reveal the underlying structural features across different scales…” However, this claim needs more precise justification. The manuscript should clearly specify which aspects of existing studies are insufficient and provide examples from the literature to support these claims.

• Alternative Approaches in Recent Studies: Other studies have used alternative methods to analyze multiscale reef structural complexity, such as: Controlling the VRM (Vector Ruggedness Measure) cell size to capture spatial variability, Computing surface rugosity while adjusting DEM resolution, Using alternative algorithms (e.g., fractal analysis). These methods also enable multiscale complexity analysis and should be acknowledged in the introduction. In Lines 74-76 and 67-69, the cited studies have already analyzed multiscale structural complexity. In the discussion section, you should describe how their indices differ from yours and clarify the distinctions between their approaches and the wavelet-based method proposed in this study.

Minor comment

1. Lines 59-70: It is unclear why this paragraph first describes the challenges of photogrammetry surveys and then concludes that UWP is a highly accurate and suitable method for modern reef assessment. Additionally, Reference [18] (a 2001 study) is outdated for discussing the challenges of 3D photogrammetry. In contrast, References [15]-[17] (published after 2015) describe how advancements in technology have made reef structural complexity analysis increasingly accessible. The literature review should be reorganized chronologically to maintain consistency in the argument.

2. The figure numbering in the text does not match the figure numbering at the end of the document (Figures 3-6). The figure captions should be placed alongside the corresponding figures, and statistical significance symbols should be clearly indicated in all figures where relevant.

3. Citations (e.g., Lines 75, 250-251): The author-year citations lack corresponding reference numbers, making it difficult to locate the sources. Ensure that all references are properly numbered.

4. In all figures, the Y-axis should explicitly indicate whether the rugosity values represent the chain method, DGR, or DLR to avoid ambiguity.

5. Line 80: Clearly define what is meant by “high-resolution coral reef elevation profiles” in terms of millimeter or centimeter scale precision.

6. Line 125: Specify which coordinate system was used in the study.

7. DGR Calculation (Lines 203-204): The manuscript states that DGR was only calculated for areas beneath live coral cover. However, this raises concerns—can DGR still effectively represent broader large-scale topographic structures if it excludes areas without live coral?

8. Results (Lines 218-221): The content about coral morphological composition is missing a reference to Figure 7, which should be explicitly mentioned for clarity.

9. Lines 227-230: The statement "Underwater digital photogrammetry improves the assessment of coral reef features by allowing more accurate quantification..." contradicts Lines 59-62, where the manuscript discusses the difficulties of 3D imaging techniques. The argument should be clarified to ensure consistency.

10. Lines 246-248: Provide specific examples supporting the relationship between multiscale structural complexity and ecological processes to strengthen this claim.

11. Lines 278-280: Instead of discussing the relationship between structural complexity and biodiversity, the manuscript should focus more on the ecological significance of different rugosity scales (DGR vs. DLR). Since the study’s core contribution lies in distinguishing structural complexity at different scales, it is essential to clarify what ecological insights each metric provides.

12. Lines 289-290: High structural complexity does not always correlate positively with biodiversity (REF). Additionally, this study does not analyze the relationship between structural complexity and biodiversity, so this statement should not be used as a concluding highlight.

13. Line 151: While the manuscript justifies using 5 cm as the minimum scale for rugosity calculations based on coral morphology, this methodological constraint does not necessarily translate into ecological relevance. Many coral colonies and reef microstructures are smaller than 5 cm, meaning that the study may overlook fine-scale complexity. The discussion section should explicitly acknowledge this limitation and its potential impact on rugosity interpretation.

**Do you want your identity to be public for this peer review?** For information about this choice, including consent withdrawal, please see our Privacy Policy

Reviewer #1: No

Reviewer #2: **Yes: ** Guanyan Keelung Chen

---

## [Author Response · Author response to Decision Letter 1]

21 Apr 2025

Response to Reviewers

Dear reviewers, we appreciate your comments and below you will find a detailed answer addressing all the suggestions.

Reviewer 1

Overall

1.- Some references are in a different format (numbered vs. spelled out), but maybe this is the journal instructions. They should be validated if they are appropriate.

Thank you for your observations. We have revised the format throughout the manuscript to ensure consistency.

2.- Introduction Section

The introduction seems to miss the existing literature on this topic. I recommend more thorough literature search on the topic of coral reef photogrammetry and rugosity. In particular, see Carlot et al 2020 (https://doi.org/10.1007/s00338-020-01916-8); Fukunaga et al 2019 (https://doi.org/10.3390/jmse7020027); Fukunaga et al 2020 (https://doi.org/10.3390/rs12061011); Fukunaga & Burns 2020 (https://doi.org/10.3390/rs12172676) to list a few. It is acknowledged this is not a review paper, but having an updated background pertinent to your topic is useful in pushing this field further.

We have updated the introduction, including current literature and its application as suggested, lines 72-87.

3.-M&M Section

Photogrammetric processing, colony identification and digitization

General – More information should be included on the Metashape parameters used to create the orthomosaics and DEMs. What were the alignment parameters and error reduction protocols used to generate the outputs? Additional, what is the accuracy of the 60x60 cm PVC markers used to scale the plots? Markers used for photogrammetry tend to be high resolution in order to provide appropriate level accuracy of the constructed models. What is the resolution of the models, what resolution were the orthomosaics, and DEMs exported. The stated protocol was a 2m altitude which is a bit high for high resolution, but helps cover larger areas, but also tends to decrease the output resolution. All this information should be incorporated somewhere in the manuscript. Also, see the Metashape report metrics which also provides additional contextual information on the models.

I recommend you provide the parameters and report metrics as well as the root mean square error of all the models as supplementary files. This provides a background on the models generated and their outputs.

We have added more information related to the photogrammetric processing, including the alignment parameters, the accuracy of the models and the spatial resolution obtained, lines 135-136, 139-140. We also have included the parameters and report metrics from the processing in the supplementary file S1 as suggested.

4.- Assessment of Rugosity.

Line 145 – acronym is incorrect, should be DLR

General – This seems to be a fairly novel approach in assessing photogrammetry models developed in other disciplines. As stated in my comments about missing relevant research in looking at similar questions about coral reefs and community structure, I suggest you incorporate some comparison analyses at some level in order to help support the use of this novel approach. It may be appropriate, but without a comparison to existing published methods, it might provide similar or different results to these methods. This approach does appear to be different by separating out local vs global rugosity, so this is certainly worth pursuing and it well supported. I recommend additional methodical description on the MODWT and Daubechies method used here as it appears to be a novel application. More information on the decomposition levels also seems warranted. Why do the first four levels represent the local features and the final level the global features?

The acronym was corrected, line 176.

As suggested, we compare our approach to other UWP-derived DEMs approaches, lines 281-286; 303-309.

We enhance discussion by incorporating comparisons with other fields and reef studies, as detailed in lines 281-286.

We have expanded the methodological description for the rugosity assessment, lines 156-170.

The wavelet analysis used decomposes a signal (in this case the height from the DEMs) considering the frequency of the data at different spatial scales (levels). On this type of analyses four levels are commonly used (Li et al 2012; Duranta et al 2023) and in our data, using more levels resulted in very small variations which can’t be represented with the obtained spatial resolution of the DEMs. The final level is the original data minus the four levels of variation.

5.- Results Section

Line 220 – should be mound, not mount?

General – See above comments on additional supplemental material.

I am having a hard time going between the results section with the figure captions and the figure themselves. There appears to be some figure reference mistakes as well. I recommend to rework this section and double check figure references. I would be interested in more in-situ vs digital rugosity comparisons.

The other aspect I realize is that I am uncertain of the rugosity metric from the digitized method. I presume it is not the same as the chain ratio method, but what units are this metric in? Unitless? Conducting a digital measurement of rugosity that is similar to the chain method is very interesting. This is a study I have been interested in and this is the first attempt at it. I do recommend flushing that out as much as possible. Even if the two metrics are different, it could be interesting to run a regression on the methods to show the correlation.

We have corrected the error in the word "mound” line 256 and included data on the resolution, RMSE and scales in the supplementary materials as previously described.

As also suggested by reviewer two, we have made changes to figures 2, 3 and 4, and reworked the results section. We have also checked for reference mistakes and corrected them.

The digital rugosity, encompassing both DLR and DGR, is quantified using Ra (equation 2), which employs the definite integral of the surface profile height over an evaluation length of 5 cm (lines 181-183).

This metric is dimensionless. We have clarified in the figures whether the rugosity assessment pertains to the chain methods, DLR, and/or DGR.

The DLR is not equivalent to the chain method. As suggested, we performed a regression between the DGR and the in situ rugosity. The data is not correlated (R2= 0.14, p=0.4613), lines 232-233.

6.- Discussion

See comments from the introduction to incorporate the newest literature on this topic. Your discuss is good, but seems to lack an in depth background on the topic. These results, while important, are not new in the global sense. They are worthy of publication, but also should incorporate a discussion of these results from results of other published work based on the wealth of background literature that exists.

We have updated the Discussion considering the comments, lines 299-309.

Reviewer 2

1.-The limitations of the chain method and multiscale structural complexity were discussed as early as 2012 (Friedman et al., 2012), and thus, this is not a novel finding. The main contribution of this study lies in applying a wavelet-based method to analyze rugosity. However, the methodological approach and visualizations do not clearly illustrate the differences between the wavelet-based method and the chain method.

Thank you for your comments. We appreciate the detailed feedback on Friedman et al. (2012) and the emphasis on the importance of multiscale rugosity measurement. Our research primarily focuses on exploring digital signature topography and decomposing rugosity into local and historical components. We have updated the introduction and discussion in order to better define the contribution of our study, lines 72-87; 299-309.

We have also improved the results section following the suggestions from both reviewers.

2.- Methodology (Lines 90-96): Since the results primarily rely on comparisons across different reefs, it is essential to provide a basic description of the 6 reef sites, especially regarding factors that influence structural complexity, such as the anthropogenic activities, historical thermal stress, and high range. This would enhance the interpretation of the rugosity differences across sites.

We have included the requested information in the study area section of the manuscript, lines the description of the study area, lines 112-118.

3.-Discussion (Lines 235-242): This section should provide a more detailed discussion on how the newly introduced rugosity metrics (DLR and DGR) differ from chain method. Specifically, what aspects of biological or abiotic reef structure do these differences reflect?

We improved the discussion in lines 299-309.

4.-Ecological Relevance (Lines 243-246): Given that the study analyzed coral cover, it would be beneficial to present the relationship between DLR, DGR, and coral cover. This would establish a stronger ecological connection between the proposed rugosity metrics and coral reefs, and allow for comparisons with other existing structural complexity metrics.

Thank you for your comment, we updated the results in lines 260-275 and have also included a figure as a supplementary file (S9 File).

5.- Figures and Statistical Consistency (Figures 3-6): I recommend merging Figures 3-6 to facilitate a direct comparison between the wavelet-based method and the chain method. The statistical approaches and labeling should also be consistent across figures. According to Line 194, Figure 3 presents median values, while other figures use mean values. The rationale for this inconsistency is unclear and should be justified or standardized.

We have merged Figures 3 and 4 as suggested. For figures 5 and 6, we believe it is better to show them separately as they have high differences in the values on the Y axis that affect visualization. We have checked for consistency on the labeling and the statistical approach.

This was a mistake and has been corrected, all values (where applicable) corresponded to the median.

6.- The explanation of the rugosity calculation method is insufficient, and the ecological significance of different rugosity scales is not well supported by the results:

• Methodology (Lines 117-118): The quality of photogrammetry images and the resolution of the DEMs significantly impact structural complexity measurements. The manuscript should provide the number of images used and the resolution of the output DEM. Additionally, it would be beneficial to include the parameter settings used in Agisoft Metashape as supplementary material to allow other researchers to compare methodologies.

We have expanded the description about the assessment of rugosity as detailed in lines 157-171. We have also incorporated the ecological significance of the different scales used and supporting data in lines 260-275 (results); 337-363 (discussion).

We have incorporated additional information on the methodology regarding the photogrammetric processing, colony identification, and digitization in lines 135-143 and included the alignment metrics as supplementary material (S1 file).

7.- Results (Figures 2a, 2b): Presenting all reefs together makes it difficult to observe finer details. I strongly suggest displaying only representative reef sites to enhance clarity. Furthermore, corresponding orthomosaics and DEMs should be included to allow for a direct comparison between DLR, DGR values, and actual ecological features.

We have revised Figure 2 as requested and have also included the remaining reefs as supplementary data in the S3 file.

8.- Statistical Interpretation (Figure 7): The significant differences between coral morphological groups in Figure 7 only indicate that DLR can distinguish between Agariciid, branching corals, and other coral growth forms, but this does not necessarily mean they are the primary contributors to reef structural complexity. Since coral morphology data were analyzed in ArcMap, the study should provide the composition of coral growth forms and coral cover at each reef site to better understand which growth forms contribute the most to structural complexity.

We respectfully disagree. We tested the contribution values to DLR by coral morphological groups with a non-parametric test (Kruskall-Wallis test, lines 202-210). The results indicate that the contribution to DLR from this groups was only different for Agariciid and Branching corals, where Agariciid corals had the higher median value. Therefore, for scleractinian corals in the reefs evaluated, Agariciids are the primary contributors to DLR. This is further supported by the correlation analysis that is now included in the manuscript, lines 260-266.

According to Atlantic Gulf Rapid Reef Assessment, Agariciid corals are composed by the following species Agaricia agaricites, Agaricia fragilis, Agaricia humilis, Agaricia lamarcki, Agaricia tenuifolia and Helioseris cucullata, that we observed in our data. The composition of coral growth forms (morphological groups) is included in the S2 file.

As suggested we now include information of coral cover and growth forms for each site, lines 269-271, Fig 8.

9.- Quantifying Coral Contributions (Line 268): When discussing “…Agariciid corals, which currently contribute the most to DLR,…” it would be more rigorous to use a Linear Model with rugosity log-transformed to quantitatively assess the contribution of different coral growth forms to DLR rather than relying solely on descriptive statistics.

As mentioned in the previous comment, we used a non-parametric test to assess the differences in contribution of morphological groups to DLR, which are not considered as descriptive statistics. The Kruskal-Wallis test compares the medians of three or more groups to test if they come from the same distribution. It is the nonparametric equivalent to a one-way ANOVA (Zar, 1984).

However, based on your valuable observations, we have enhanced this section by adding figures that visualize coral cover versus DLR for each group and a linear correlation analysis was also performed as suggested, to assess the contribution of the different coral growth forms to DLR, lines 251-275, and fig 7.

10.- Missing Data on Coral Cover Relationships (Lines 264-267): The manuscript frequently discusses the relationship between DLR, DGR, and coral cover, but there is a lack of supporting visual data. Although the methodology mentions such an analysis, no figures or correlation analyses are provided. I recommend including scatter plots of DLR versus coral cover at each plot and conducting a correlation analysis.

We tried using scatterplots and correlation analysis as suggested. However, we did not identify a distinct correlation for each reef (R2 values for linear regressions were very low), as illustrated in the figures included in the answer to reviewers file. However, as mentioned in the previous comment, we added figure 7 and evaluated the linear correlation between total coral cover for both DGR and DLR, and the contribution of the different growth forms: S9 file, fig 7 and lines 260-271. We have also included a graph showing the coral cover combined and separated by site, fig 8, lines 269-275.

11.-Supporting Evidence for Yucab Reef Example (Lines 259-261, 274-277): The manuscript repeatedly states that DLR reflects biological contributions from coral growth forms, while DGR represents the broader topographic variability shaped by historical reef development, using Yucab Reef as an example of high DGR and low DLR. While this is a valuable insight, it requires additional supporting literature or actual reef site photographs to substantiate the claim.

We have provided supporting literature as suggested, lines 327-331.

12.- The manuscript lacks comparisons with recent studies (post-2015) on reef structural complexity, making it difficult to clearly assess the advantages and limitations of the wavelet-based method relative to other approaches. Without these comparisons, the study does not fully demonstrate how the wavelet-based method differs from or improves upon existing techniques:

• Introduction (Lines 72-74): The manuscript states, “…Existing methods neither adequately capture the spatial variability of reef complexity nor effectively decompose topographic signatures to reveal the underlying structural featu

---

## [Decision Letter · Decision Letter 1]

Dear Dr. Rioja-Nieto,

Thank you for submitting your manuscript to PLOS ONE. After careful consideration, we feel that it has merit but does not fully meet PLOS ONE’s publication criteria as it currently stands. Therefore, we invite you to submit a revised version of the manuscript that addresses the points raised during the review process.

We look forward to receiving your revised manuscript.

Kind regards,

Parviz Tavakoli-Kolour

Academic Editor

PLOS ONE

Journal Requirements:

Reviewers' comments:

Reviewer's Responses to Questions

**Comments to the Author**

Reviewer #2: (No Response)

2. Is the manuscript technically sound, and do the data support the conclusions?

Reviewer #2: Partly

3. Has the statistical analysis been performed appropriately and rigorously?

Reviewer #2: Yes

4. Have the authors made all data underlying the findings in their manuscript fully available?

Reviewer #2: Yes

5. Is the manuscript presented in an intelligible fashion and written in standard English?

Reviewer #2: Yes

Reviewer #2: I would like to thank the authors for the comprehensive revision, and explanation of all changes made. The manuscript has improved significantly in important aspects. However, the following issues still need to be addressed:

Please define “rugosity” more clearly. In some passages you might consider replacing “rugosity” with the broader term like “reef structural complexity” (e.g., lines 18, 35, 153, 299, 319, 370…). In most of the literature, rugosity is defined simply as the ratio of actual surface length to geometric length. And it can be obtained by different methods, for example the chain method, using a depth gauge, or derived from a DEM or point cloud. But some of the metrics you list, such as fractal dimension and VRM mentioned on lines 304-305, are not strictly rugosity. Because these structural complexity metrics are calculated differently, they are not “rugosity metrics.” Therefore, please revise some of the terminology to avoid confusion. Also, calling your new metrics “rugosity” may lead readers to assume they are based on traditional measurement methods, but since you use wavelet-based approach, DLR and DGR represent a different concept. Could you consider giving them new names to avoid confusion with past metrics? Alternatively, please explain why it is appropriate to include the term “rugosity” in DLR and DGR.

Line 33-34. “…Our findings indicate that conventional approaches for estimating rugosity undervalue the historical role on assessments and…” Since there are many methods for analyzing reef structural complexity and this study only compared against the chain method, I suggest replacing “conventional approaches” with “the chain method” and replacing “rugosity” to “structural complexity” to better reflect the actual results.

Although the computational principles of your approach differ, if you claim that other methods cannot achieve the same results, it is necessary to analyze more of the previous structural complexity metrics. The paper currently only compares against the earliest chain method, lacks concrete evidence of the shortcomings of other past approaches (as mentioned in lines 79–82). For example, analyses using varied DEM resolutions and different structural complexity metrics can likewise distinguish living biological tissue from the topographic features (see Fukunaga & Burns, 2020; Engleman et al., 2023; Harris et al., 2023). You should include more discussion comparing your method to previous ones; otherwise, the statements in lines 33-34, 299-301, 307-309, 369-371 come across as overinterpretation and should be removed or rephrased.

Fukunaga, A., & Burns, J. H. R. (2020). Metrics of Coral Reef Structural Complexity Extracted from 3D Mesh Models and Digital Elevation Models. Remote Sensing, 12(17), 2676. https://doi.org/10.3390/rs12172676

Engleman A, Cox K, Brooke S. (2023). Dead but not forgotten: complexity of Acropora palmata colonies increases with greater composition of dead coral. PeerJ 11:e16101 https://doi.org/10.7717/peerj.16101

Harris, D. L., Webster, J. M., Vila-Concejo, A., Duce, S., Leon, J. X., & Hacker, J. (2023). Defining multi-scale surface roughness of a coral reef using a high-resolution LiDAR digital elevation model. Geomorphology, 439, 108852.

Figure 2, There is still a lot of empty white space around the figure. Could you adjust the layout or crop the image to show only the relevant area so that the graphic fills more of the space and appears larger and clearer?

Figure 6, the grouping categories should be made consistent. For example, “Agariciid” refers to a taxonomic family, whereas “Branching” is a growth form, mixing taxonomic and morphological categories is not academically rigorous. I recommend changing “Agariciid” to the corresponding growth form to maintain consistency, using a scheme like the morphological classification in Zawada et al. (2019). Additionally, it is unclear what “flower” means in “Flower & Solitary”; please replace it with a term that is more commonly used in the literature.

Zawada, K. J., Dornelas, M., & Madin, J. S. (2019). Quantifying coral morphology. Coral Reefs, 38(6), 1281-1292.

Figure 6, for the three groups on the right that are neither A nor B, please add the label C to indicate that they do not differ significantly from each other but are different from A and B.

For Figure 7, the sample size at each site is very small. If you do not plan to discuss the site‐specific relationship between DLR and coral cover in the content, you could plot all data together to show the overall correlation between DLR and coral cover (and, if needed, use different colors for points from each site). Also, I may have missed it, but Figure 7 is not cited anywhere in the manuscript, please refer to it in the appropriate paragraph.

For Figure 8, I recommend using a stacked percentage bar chart to more clearly illustrate the composition of each site and the overall group breakdown. Listing individual species in the figure is somewhat distracting; you can instead describe in the text which species belong to each group.

**Do you want your identity to be public for this peer review?** For information about this choice, including consent withdrawal, please see our Privacy Policy

Reviewer #2: **Yes: ** Guan-Yan Chen

---

## [Author Response · Author response to Decision Letter 2]

4 Jul 2025

Response to Reviewers

Dear reviewer, we appreciate your comments and below you will find a detailed answer addressing all the suggestions.

Reviewer 2.

Reviewer #2: I would like to thank the authors for the comprehensive revision, and explanation of all changes made. The manuscript has improved significantly in important aspects. However, the following issues still need to be addressed:

Please define “rugosity” more clearly. In some passages you might consider replacing “rugosity” with the broader term like “reef structural complexity” (e.g., lines 18, 35, 153, 299, 319, 370…). In most of the literature, rugosity is defined simply as the ratio of actual surface length to geometric length. And it can be obtained by different methods, for example the chain method, using a depth gauge, or derived from a DEM or point cloud. But some of the metrics you list, such as fractal dimension and VRM mentioned on lines 304-305, are not strictly rugosity. Because these structural complexity metrics are calculated differently, they are not “rugosity metrics.” Therefore, please revise some of the terminology to avoid confusion. Also, calling your new metrics “rugosity” may lead readers to assume they are based on traditional measurement methods, but since you use wavelet-based approach, DLR and DGR represent a different concept. Could you consider giving them new names to avoid confusion with past metrics? Alternatively, please explain why it is appropriate to include the term “rugosity” in DLR and DGR.

Thank you for the suggestions, we agree and have categorized the metrics based on the nature of the calculations to enhance clarity and prevent confusion, lines 73-81. We have also replaced where appliable the term rugosity with either reef structural complexity, structural complexity or complexity. This included modifying the title of the manuscript. Finally, as suggested we have also changed the names of DLR and DGR to Digital Local Complexity (DLC) and Digital Global Complexity (DGC) to describe the two components in our approach.

Line 33-34. “…Our findings indicate that conventional approaches for estimating rugosity undervalue the historical role on assessments and…” Since there are many methods for analyzing reef structural complexity and this study only compared against the chain method, I suggest replacing “conventional approaches” with “the chain method” and replacing “rugosity” to “structural complexity” to better reflect the actual results.

Change performed lines 34-36.

Although the computational principles of your approach differ, if you claim that other methods cannot achieve the same results, it is necessary to analyze more of the previous structural complexity metrics. The paper currently only compares against the earliest chain method, lacks concrete evidence of the shortcomings of other past approaches (as mentioned in lines 79–82). For example, analyses using varied DEM resolutions and different structural complexity metrics can likewise distinguish living biological tissue from the topographic features (see Fukunaga & Burns, 2020; Engleman et al., 2023; Harris et al., 2023). You should include more discussion comparing your method to previous ones; otherwise, the statements in lines 33-34, 299-301, 307-309, 369-371 come across as overinterpretation and should be removed or rephrased.

Fukunaga, A., & Burns, J. H. R. (2020). Metrics of Coral Reef Structural Complexity Extracted from 3D Mesh Models and Digital Elevation Models. Remote Sensing, 12(17), 2676. https://doi.org/10.3390/rs12172676

Engleman A, Cox K, Brooke S. (2023). Dead but not forgotten: complexity of Acropora palmata colonies increases with greater composition of dead coral. PeerJ 11:e16101 https://doi.org/10.7717/peerj.16101

Harris, D. L., Webster, J. M., Vila-Concejo, A., Duce, S., Leon, J. X., & Hacker, J. (2023). Defining multi-scale surface roughness of a coral reef using a high-resolution LiDAR digital elevation model. Geomorphology, 439, 108852.

We revised and modified the text to better explain the main distinctions between our analysis and other methodologies, such as Vector Roughness Measurement and variation in Digital Elevation Model resolutions, in lines 73-81, 85-89. Furthermore, we improved the discussion in lines 318-325; changed as suggested in a previous comment the lines 34-36, and revised statements in lines 312-313; 386-387.

Figure 2, There is still a lot of empty white space around the figure. Could you adjust the layout or crop the image to show only the relevant area so that the graphic fills more of the space and appears larger and clearer?

We changed the figure 2 by increasing the zoom level.

Figure 6, the grouping categories should be made consistent. For example, “Agariciid” refers to a taxonomic family, whereas “Branching” is a growth form, mixing taxonomic and morphological categories is not academically rigorous. I recommend changing “Agariciid” to the corresponding growth form to maintain consistency, using a scheme like the morphological classification in Zawada et al. (2019). Additionally, it is unclear what “flower” means in “Flower & Solitary”; please replace it with a term that is more commonly used in the literature.

Zawada, K. J., Dornelas, M., & Madin, J. S. (2019). Quantifying coral morphology. Coral Reefs, 38(6), 1281-1292.

We appreciate the suggestions. Although there are alternative morphological categories; we have adopted the grouping categories established by the Atlantic and Gulf Rapid Reef Assessment (AGRRA) initiative [1] lines 157-158. This framework aims to standardize data collection across the region, thereby enhancing the comparability of datasets. We employed these categories to facilitate regional comparisons.

Figure 6, for the three groups on the right that are neither A nor B, please add the label C to indicate that they do not differ significantly from each other but are different from A and B.

We updated the figure as suggested.

For Figure 7, the sample size at each site is very small. If you do not plan to discuss the site‐specific relationship between DLR and coral cover in the content, you could plot all data together to show the overall correlation between DLR and coral cover (and, if needed, use different colors for points from each site). Also, I may have missed it, but Figure 7 is not cited anywhere in the manuscript, please refer to it in the appropriate paragraph.

This figure shows the DLC and coral coverage grouped by morphological groups (fig 7a-e). When grouping the data together, the figure becomes difficult to understand. Therefore, we prefer to maintain it as it is, as it was suggested by the other reviewer. This figure is described in lines 272-280.

For Figure 8, I recommend using a stacked percentage bar chart to more clearly illustrate the composition of each site and the overall group breakdown. Listing individual species in the figure is somewhat distracting; you can instead describe in the text which species belong to each group.

We have changed the figure as recommended and have included a reference to the supplementary information.

1. Lang JC, Marks KW, Kramer PR, Kramer PA, Ginsburg RN. Protocolos AGRRA version 5.5. Big Pine Key: Ocean Research & Education. 2012.

---

## [Editor Report · Decision Letter 2]

Multiscale Structural Complexity Assessment of Coral Reefs Using Underwater Photogrammetry

PONE-D-25-02577R2

Dear Dr. Rioja-Nieto,

We’re pleased to inform you that your manuscript has been judged scientifically suitable for publication and will be formally accepted for publication once it meets all outstanding technical requirements.

Kind regards,

Parviz Tavakoli-Kolour

Academic Editor

PLOS ONE